# Mechanism of Citri Reticulatae Pericarpium as an Anticancer Agent from the Perspective of Flavonoids: A Review

**DOI:** 10.3390/molecules27175622

**Published:** 2022-08-31

**Authors:** Li Song, Peiyu Xiong, Wei Zhang, Hengchang Hu, Songqi Tang, Bo Jia, Wei Huang

**Affiliations:** 1College of Basic Medicine, Chengdu University of Traditional Chinese Medicine, Chengdu 610000, China; 2College of Traditional Chinese Medicine, Hainan Medical University, Haikou 571199, China

**Keywords:** Citri Reticulatae Pericarpium, flavonoids, anticancer, mechanism, phenotype

## Abstract

Citri Reticulatae Pericarpium (CRP), also known as “chenpi”, is the most common qi-regulating drug in traditional Chinese medicine. It is often used to treat cough and indigestion, but in recent years, it has been found to have multi-faceted anti-cancer effects. This article reviews the pharmacology of CRP and the mechanism of the action of flavonoids, the key components of CRP, against cancers including breast cancer, lung cancer, prostate cancer, hepatic carcinoma, gastric cancer, colorectal cancer, esophageal cancer, cervical cancer, bladder cancer and other cancers with a high diagnosis rate. Finally, the specific roles of CRP in important phenotypes such as cell proliferation, apoptosis, autophagy and migration–invasion in cancer were analyzed, and the possible prospects and deficiencies of CRP as an anticancer agent were evaluated.

## 1. Introduction 

Citri Reticulatae Pericarpium is a commonly used traditional Chinese medicine derived from the ripe peel of the Rutaceae plant *Citrus reticulate Blanco* and its cultivars [1], which was first recorded in *Shen Nong Ben Cao Jing* and has a history of thousands of years in China. As a botanical medicine with the same origin of medicine and food, CRP has various pharmacological effects, which can be used alone or combined with other traditional Chinese medicines to form many well-known classical prescriptions. It is widely used in the clinical treatment of diseases of various systems and has outstanding advantages for diseases of the respiratory and digestive system, especially diseases with cough, expectoration, nausea and vomiting as the main symptoms.

The clinical efficacy of CRP has been affirmed, especially its contribution in cancer treatment. Researchers began to explore its mechanism, trying to find out its countermeasures against cancer in cancer cell lines and animal models, and concluded that the natural compounds flavonoids contained in CRP are the implementers of its anti-cancer effect. According to reports, flavonoids of CRP have outstanding performance in terms of regulating key signaling pathways and related effectors, blocking the cancer cell cycle to resist proliferation, inducing cell apoptosis, enhancing autophagy and inhibiting cell migration and invasion. Thanks to these research results and data, the core code of CRP as an anticancer agent has gradually been revealed.

As a novel anticancer agent, CRP has received extensive attention, and many studies have confirmed that CRP and its active ingredients have inhibitory effects on cancer. This review aims to collect and introduce the mechanism of CRP and its components in inhibiting cancer, focusing on breast cancer, lung cancer, prostate cancer, hepatic carcinoma, gastric cancer, colorectal cancer, esophageal cancer, cervical cancer, bladder cancer and other cancers with a high diagnosis rate.

## 2. Pharmacological Effects and Chemical Composition of CRP

Pharmacological studies have found that when applied to the digestive system, the efficacy of CRP is to ameliorate gastrointestinal smooth muscle activity, accelerate gastric emptying and intestinal push, alter gut microbiota, protect the esophagus and gastrointestinal mucosa and resist peptic ulcer [2,3,4,5,6,7]. To date, CRP has proven effects in the respiratory system, including but not limited to inhibiting airway inflammation, fighting acute lung injury and pulmonary fibrosis [8,9,10,11,12,13]. In terms of the cardiovascular system, CRP can improve cardiac insufficiency, alleviate cardiac hypertrophy and myocardial fibrosis, prevent hyperlipidemia and keep cardiomyocytes away from the damage of hypercholesterolemia [14,15,16,17,18]. Experiments have found that CRP’s neuroprotective properties confer the ability to prevent neurodegeneration caused by Parkinson’s, Alzheimer’s, Huntington’s disease and multiple sclerosis [19,20,21,22]. Unlike the above systems, CRP has relatively few studies in the related direction of the urinary system and has been reported to be anti-inflammatory and enhance renal function [23,24,25]. It is worth mentioning that CRP exhibits anti-inflammatory activity not only in the urinary system, but also in various systems of the body. Nobiletin is considered to be a marker of anti-inflammatory effect of CRP, while Xue N. et al. suggested that naringenin inhibits reactive oxygen species (ROS) and inflammatory cytokines involved in the process of CRP against inflammatory damage [26,27]. The antioxidant capacity of CRP is also an important part of its pharmacological action, and it is positively correlated with the aging of CRP [28]. Data proves that long-term storage benefits the quality of CRP, in which phenolic acids play a key role [29]. Moreover, the antioxidant activity of CRP is closely related to the fungi represented by Aspergillus niger growing on its surface, which can promote the transformation of flavonoids in CRP to increase its content [30].

The composition of natural compounds of CRP is complex. Through supercritical CO_2_ extraction [31,32], Soxhlet extraction [1,33,34], pressurized liquid extraction (Wei L) and high-speed countercurrent separation [35], 161 constituents from CRP have been extracted and identified, including 65 flavonoids, 51 phenolic acids, 27 fatty acids and 18 amino acids [29,36]. Certainly, flavonoids are the most relevant class of all compounds with the pharmacological effects of CRP, and flavanone, flavone, and polymethoxyflavone aglycones, flavanone-and flavone-O-glycosides and flavone-C-glycosides are all the compounds it contains [37,38,39]. Due to its abundant content, hesperidin has been used as a chemical reference for CRP quality control by the *Chinese Pharmacopoeia*. Among them, in terms of exerting medical effects, the representative flavonoids are tangeretin and nobiletin, belonging to the polymethoxyflavones, and naringin and hesperidin, belonging to the flavanone O-glacosides (Table 1, Figure 1 and Figure 2).

## 3. Epidemiological Investigation of Cancer

Cancer develops from the clonal expansion of abnormal cells inside the body [41]. As changes in the prevalence and distribution of the main risk factors, cancer incidence and mortality are rapidly growing worldwide. Cancer was the leading cause of premature death in 57 countries until 2020 [42]. Based on cancer incidence and mortality from the GLOBOCAN 2020 and The World Health Organization database, the most commonly diagnosed cancers worldwide were female breast cancer, lung and prostate cancer; the most common cause of cancer death were lung, liver and stomach cancers [43,44]. Female breast cancer has now surpassed lung cancer as the leading cause of global cancer incidence in 2020 with more than 6 million deaths [45]. The causes of cancer are multifaceted, with multiple external factors combined with internal genetic changes leading to cancer, but it mainly originates from both environment and genetics [46]. Unhealthy lifestyles, such as cigarette smoking, alcohol abuse, excess fat and red meat intake and lack of fiber, can also affect the development of cancer [47,48]. Conventional treatment modalities for cancer include surgery, radiation therapy, chemotherapy, targeted therapy, hormonal therapy and immunotherapy [49]. However, current treatments are often accompanied by various physical and psychological side effects, which severely impact the prognosis and life expectancy of patients. Throughout the past few years, both clinical and laboratory studies of the treatment of cancer through traditional Chinese medicine have gained great attention. The efficacy and safety of herbal medicine make it unique in the treatment of cancer. At the same time, Chinese medicine can also be used as an adjuvant to reduce the side effects of conventional cancer treatment.

## 4. The Performance of CRP in the Typical Phenotype of Cancer

Broadly speaking, cancer is a disease caused by dysregulation of cell proliferation [50]. The manifestation of cell proliferation is cell division, and mitosis is the main means of cell division in eukaryotes, which is periodic. The cell cycle goes through 4 phases: G1, S, G2 and M; and 3 checkpoints: G1/S, G2/M and SAC [51]. Apparently, this program is regulated by proliferation signals, cyclins and corresponding cyclin-dependent kinases (CDKs) [52]. Inactivation or mutation of growth suppressor genes and overexpression of oncogenes lead to uncontrolled proliferation of cancer cells and upregulation of cyclin-CDKs’ expression, which in turn affects cell cycle progression and mitosis [53]. However, mTOR, which coordinates growth metabolism, suppresses the PI3K/AKT pathway to attenuate its anti-proliferative effect [54]. From the perspective of cancer metabolism, one of the biological markers of cancer, the Warburg effect, a form of energy metabolism manifested in aerobic glycolysis, enhances the proliferation and division, invasion and anti-apoptosis of cancer cells [55]. It is worth noting that in the G1 phase, on the one hand, cyclin D1 inactivates the inhibition of a mitochondria through the action of a series of factors, such as NRF-1, PPARγ and PGC-1α [56,57,58]. On the other hand, during G1 phase in cancer cells, PKM2 affected by PFK1, Ras, HIF-1 and PI3K/AKT/mTOR upregulates the gene CCND1 encoding cyclin D1 by increasing c-Myc expression and promoting β-catenin transactivation [59,60,61]. Conversely, PKM2 activated under the action of proliferative signals, such as EGFR, NF-κB and AKT, can also upregulate the expression of HIF-1, STAT3 and c-Myc through the STAT3 signaling pathway to maintain cell cycle progression [62,63,64]. Naringin and tangeretin were determined to affect cyclin D and beta-catenin, resulting in cell cycle arrest, which in turn counteracts the disordered proliferation of cancer cells. In addition, in the anti-proliferation process of CRP, effectors on JAK/STAT, PI3K/AKT/mTOR pathways are also involved (Figure 3).

Interestingly, some proteins are involved in both cell proliferation and apoptosis. For example, p53 not only acts on its key effectors PUMA and p21 to induce cell cycle arrest to inhibit proliferation, but also acts as an upstream activator of pro-apoptotic proteins to initiate apoptosis [65,66]. It is reasonable to think that the cycle arrest caused by the regulation of P53 by hesperidin and tangeretin is the result of this.

The morphological changes of apoptosis are cytoplasmic shrinkage, chromatin pyknosis, nuclear fragmentation and plasma membrane blebbing, which finally form an apoptotic body [67]. Biochemical changes that can be observed during this process include caspase activation, DNA and protein breakdown, membrane changes and recognition by phagocytes [68]. The intrinsic apoptotic pathway is due to growth factor loss, DNA damage, ER stress, ROS overload, replication stress, microtubule alterations and mitotic defects, resulting in irreversible enhancement of mitochondrial outer membrane permeability and the release of pro-apoptotic factors [67,69]. The BCL-2 family is an indispensable regulatory protein in the internal apoptosis pathway, including the pro-apoptotic BAX, BAK, BAD, BCL-XS, BID, BIK, BIM and Hrk, and the anti-apoptotic BCL-2, BCL -Xl, BCL-W, BFL-1 and MCL-1 [70]. Under the action of cytotoxic signals, BH3-like proteins, such as BIM, BAD, PUMA and Noxa, transmit the signals to the downstream BAX and BAK. Under normal conditions, BAX located in the cytoplasm is enriched in mitochondria and becomes an activated state [71,72,73]. During this process, BAK is also activated. The apoptosis-inhibiting proteins BCL-2, BCL-XL, MCL-1, etc., can combine with BAX and BAK to suppress their activity [74]. Cytochrome *c* is released under the action of these two diametrically opposed proteins and then binds to Apaf-1 under the induction of apoptotic factors, such as Smac, DIABLO and Omi/HtrA2, to form apoptosome to further activate caspase-3 [75]. The apoptosis inhibitor IAP family binds to and prevents the activation of caspase, forming a negative feedback mechanism [76]. Another apoptosis-inducing factor induces apoptosis in a caspase-independent manner. Excessive DNA damage leads to the onset of the PPAR-1-dependent cell death program; AIF then translocates from the mitochondria to the nucleus, followed by nuclear condensation, phosphatidylserine exposure at the plasma membrane and mitochondrial transmembrane collapse, a process unaffected by caspase inhibitors [77]. The extrinsic apoptosis pathway is completed by death receptors that are members of the TNFR superfamily, and the death receptors that receive cytotoxic signals bind to their ligands. For instance, Fas/FasL, the most representative one, activates pro-caspase-8 into caspase-8 through the combination of FADD and DED and then activates other executioners of the caspase family in turn [78,79,80]. Activated caspase-8 activates BAX and BAK by cleaving BID to release cytochrome c, a link that links the extrinsic and intrinsic apoptotic pathways. Activated caspase-3 and caspase-7 cleave ICAD, which is an inhibitor of caspase-activated DNase, and then release CAD, resulting in DNA fragmentation and ultimately, complete apoptosis [78]. CRP is involved in the release of related proteins on the internal/external apoptotic pathway, as well as upstream and downstream signaling pathways, including but not limited to ERK and PPARγ-dependent or independent (Figure 4).

Autophagy suppresses tumors by eliminating oncogenic substrate proteins, toxic unfolded proteins and damaged organelles, preventing chronic tissue damage and cancer development; conversely, autophagy-mediated intracellular recycling promotes tumor growth metabolism in cancer [81]. Autophagy was originally thought to be tumor-suppressive, as the absence of the autophagy-related protein ATG6 has been observed in human breast, ovarian and other cancers [82]. The mTOR and ULK complexes mediated by AMPK, p53, PI3K/AKT and MAPK/ERK cooperate with ATG on the autophagic membrane to form autophagosomes, which in turn control the entire autophagy process [83,84,85,86]. The formation of autophagosomes involves a series of proteins and multimolecular complexes, involving BECLIN 1, LC3, Rab complex, etc. [86]. The damaged organelles are captured by the autophagosome membrane, recognized by the autophagy substrate p62, and then degraded by lysosomes [87]. Autophagy inhibition is beneficial to tumor growth, as its deficiency leads to the accumulation of p62, and the binding of p62 to mTORC1 inhibits autophagy and activates NF-κB and NRF-2, which further promotes tumor cell proliferation [88,89,90]. From a metabolic point of view, the existence of autophagy allows cells to still metabolize in a starved state, which means that the activation of autophagy is sufficient to maintain tumor cell survival [87,91]. CRP has not been researched in the field of autophagy for a long time, but it is currently certain that its effects on autophagy include but are not limited to regulating the PI3K/AKT/GSK-3c/mTOR pathway (Figure 5).

Cancer metastasis is the spread of cancer cells to distant organs, and the invasion–migration of cancer cells during this process is controlled by multiple factors. The process of cell invasion–migration consists of the following aspects: (a) local invasion and cell migration of the basement membrane; (b) intravascular deep vasculature and/or lymphatic system; (c) survival in the circulation; (d) arrest and extravasation at distant organ sites; and (e) colonization at metastatic sites [92]. Protocols for single cell invasion involve protease, actin cytoskeleton, integrin-dependent, integrin-independent and Rho- and Rock/MLCK-dependent modes of migration into mesenchymal migration and glide-like “amoematoid-migration” [93,94]. Epithelial–mesenchymal transitions (EMT) during cell migration and invasion is regulated by transcription factors such as slug, Twist, ZEB1 and ZEB2, which inhibit E-cadherin, the cornerstone of the epithelial state [95,96]. In addition, the double negative feedback loop formed by miR-200 and ZEB1/ZEB2 is another important mechanism for regulating EMT [96]. The invasion–metastasis cascade of cancer cells begins with the destruction of BM, and its damage is the result of MMPs as enzymatically hydrolyzed active proteins [97,98]. Indeed, the effects of MMPs on cancer are multifaceted, including inflammation, cell proliferation, extracellular matrix (ECM) degradation, cell migration, resistance to cell death, replicative immortality and metastatic niches; the most critical role is the degradation and remodeling of the ECM during cancer cell invasion and metastasis [99]. One of the sources of TGF-β, which has the function of inhibiting tumor cell differentiation, is the inactive precursor after proteolysis of MMP-9. At the same time, the regulation of VEGF by MMP-9 can promote tumor angiogenesis, which is beneficial to tumor colonization [100,101,102]. In addition, TGF-β1 can also be activated by MMP-14 and MMP-2 [103]. All three of the above proteins cleave LTBP-1 of the ECM to indirectly regulate the activity of TGF-β [104,105]. NF-κB enhances the production of MMPs, and its ability is restricted by TIMPs, with which it can form complexes that inhibit proteolysis [106,107]. In particular, MMP-7 inhibits apoptosis by cleaving Fas ligands, making it a place in the process of apoptosis [108]. The ability of hesperidin, naringin to downregulate MMPs has been demonstrated, which allows it to successfully reverse EMT (Figure 6).

## 5. Inhibitory Effect of CRP and Its Active Components on Cancer

### 5.1. Breast Cancer

With the highest diagnosis rate in women, breast cancer severely threatens the survival and quality of life of women around the world. The mechanism of CRP in the treatment of breast cancer is still under study. According to the existing results, flavonoids such as nobiletin, tangeretin, hesperidin and naringin play a crucial role. The study found that nobiletin can simultaneously inhibit the ERK1/2 and PI3K/AKT pathways to suppress the growth of TNBC MDA-MB-468 cells and perform anti-tumor effects through anti-proliferation and induction of apoptosis [109]. Tangeretin inhibits breast cancer cell metastasis by targeting TP53, PTGS2, MMP9 and PIK3CA and regulating the PI3K/AKT pathway [110]. Tangeretin inhibits the formation of BCSCs and targets BCSCs by inhibiting the Stat3/Sox2 signaling pathway, thereby treating breast cancer and BCSCs [111]. Tangeretin also attenuates DMBA-induced oxidative stress, reduces kidney DNA damage and has chemo preventive activity against DMBA-induced breast cancer with cellular engraftment [112,113]. Hesperidin exhibits a concentration-dependent cytotoxicity effect on human breast cancer cell line MCF-7, which induces apoptosis and causes DNA damage [114,115]. The combination of hesperidin and chlorogenic acid modulates mitochondrial and ATP production via the estrogen receptor pathway and synergistically inhibits the growth of MCF-7 [116]. Naringin inhibits cell proliferation and promotes apoptosis and G1 cycle arrest through regulating the β-catenin pathway, thereby suppressing the growth potential of TNBC cells [117]. What’s more, hesperidin performed the inhibitory activity of the proliferation of MCF-7-GFP-Tubulin cells, fought against drug-resistant cancer cells and amelioratde the cell migration of MDA-MB 231 cells [118,119,120].

### 5.2. Lung Cancer

As the second most common cancer worldwide, lung cancer is the leading cause of cancer death in 2020, with incidence and mortality rates approximately twice as high in men as in women [45]. Under the current situation of extremely low survival rate, how to improve the rate and the quality of life of patients diagnosed with lung cancer has become the main subject of medical research. CRP is good at treating respiratory system diseases, and pharmacological research has carried out in-depth exploration on the treatment of lung cancer. Smoking greatly increases the risk of lung cancer. Hesperidin is able to down-regulate the expression of MMPs and enhance antioxidant status to combat nicotine toxicity and suppression of smoking-induced lung cancer [121]. The antioxidant capacity of hesperidin also inhibits tumor cell proliferation in benzo(a)pyrene-induced lung cancer mouse models [122]. Another study found that inhibition of NSCLC cells proliferation and promotion of apoptosis through the miR-132/ZEB2 signaling pathway may be one of the mechanisms by which hesperidin alleviates NSCLC [123]. Hesperidin also induces apoptosis through the mitochondrial pathway, up-regulates the expression of P21 and P53 to triggers G0/G1 phase arrest in A549 cells and down-regulates cyclin D1 for anti-proliferation [124,125]. Another set of experiments by Xia R found that blocking the SDF-1/CXCR-4 pathway to inhibit the migration of A549 cells and the suppression of EMT phenotype transformation are also the approach for hesperidin to prevent tumors and its metastasis [126]. Taking A549 cells as the research object, another important natural compound of CRP, naringin, attenuates the EGF-induced MUC5AC mucin and mRNA overexpression by inhibiting the synergistic activity of MAPKs/AP-1 and IKKs/IκB/NF-κB signaling pathways [127]. Chen M found that naringin exhibited the capacity to inhibit PI3K/AKT/mTOR and NF-κB pathways and activate the expression of miR-126 in H69 cells, thereby preventing cell growth and inducing apoptosis in SCLC cells [128].

### 5.3. Prostate Cancer

Prostate cancer is the most common malignancy among men worldwide, with 1.4 million cases diagnosed in 2016 and more than 380,000 deaths [129]. A series of experimental studies have demonstrated that flavonoids in CRP have positive effects on weakening cell viability and inducing cytotoxicity in prostate cancer. The expressions of NF-κB and HIF-1α were down-regulated in nobiletin-treated prostate cell lines DU145 and PC-3 cells, accompanied by decreased phosphorylation of AKT, which in turn impairs cell viability [130]. Coincidentally, tangeretin also down-regulated the expression of AKT and AR in C4-2 cells and synergistically antagonized the resistance of CRPC cells to sorafenib or cisplatin [131]. In PC-3 cells, the pathways of tangeretin-induced cytotoxicity include not only caspase-30mediated apoptosis, but also inhibition of PI3K/AKT/mTOR pathway to reverse the EMT process [132]. 

### 5.4. Liver Cancer

Primary liver cancer, including hepatocellular carcinoma (HCC) and intrahepatic cholangiocarcinoma, was the sixth most commonly diagnosed cancer and the third leading cause of cancer death globally in 2020, with HCC accounting for 75–85% of these [45,133]. Hence, the exploration of CRP in the treatment of liver cancer mainly focuses on the direction of HCC. Zheng J. et al. verified that the regulation of JNK/Bcl-2/BECLIN1 pathway-mediated autophagy is the mechanism by which tangeretin antagonizes the proliferation and migration of HepG2 cells [134]. Hesperidin’ fight against HCC cells’ invasiveness is achieved by suppressing the activities of NF-κB and AP-1 to down-regulate the expression and secretion of MMP-9 in acetaldehyde- and TPA-induced HCC [135,136]. The pro-apoptotic protein BAX is the key to cell apoptosis, and up-regulation of BAX to induce apoptosis in HepG2 cells is an effective way for hesperidin and naringin to inhibit liver cancer [137,138]. The fact that naringin reduces cell proliferation in DEN-induced hepatocarcinoma rats is manifested by a marked decrease in AgNOR/nuclear and PCNA levels, as well as altered DNA fragmentation in liver tissue [139].

### 5.5. Gastric Cancer

According to statistics, in 2020, there were more than 1 million new cases of gastric cancer, causing 769,000 deaths, ranking fourth and fifth in the world in morbidity and mortality, respectively [45]. Relying on the long-term clinical experience of CRP in the treatment of digestive system diseases, related experiments have thus explored multiple mechanisms of anti-gastric cancer. Endoplasmic reticulum stress mediates apoptosis and autophagy. Nobiletin down-regulates AKT/mTOR signaling pathway to promote endoplasmic reticulum stress response in SNU-16 cells, which may be an integral part of its anticancer activity [140]. Mitochondrial apoptosis mediated by activated caspase-9 and Fas/Fas L synergistically enables tangeretin to fulfill its mission of inhibiting AGS cells [141]. Mitochondrial-mediated apoptosis is also applicable to the killing effect of hesperidin on AGS cells. The level of reactive oxygen species in AGS cells after hesperidin intervention increases, and the MAPK signaling pathway is regulated to induce cell apoptosis [142]. Radiation therapy is one of the main methods for the treatment of tumors at present, through downregulating the expression of Notch-, Jagged1/2, Hey-1 and Hes-1, downregulating the expression of miR-410, causing attenuated invasion and migration in GC cells; tangeretin greatly enhanced the radiosensitivity of GC cells [143].

### 5.6. Colorectal Cancer

As the third most common cancer in terms of incidence and the second in mortality, colorectal cancer accounts for one tenth of all diagnosed cancers and deaths [45]. Similar to gastric cancer, the advantages of CRP in digestive diseases are also reflected in colorectal cancer. Nobiletin and its major metabolites M1, M2 and M3 in the colon have established roles in cell cycle arrest and apoptosis, thus effectively inhibiting AOM/DSS induced colitis-associated colon cancer in CD-1 male mice [144]. In rectal cancer, the mechanism of nobiletin is downregulating MMP-7 gene expression to inhibit the invasion and metastasis of cancer cells [145]. Tangeretin and 5-FU synergistically up-regulate P21 in HCT-116 cells, which in turn activates the P53-mediated DNA damage response and triggers apoptosis via the JNK pathway, suggesting that the combination of tangeretin and 5-FU suppresses the autophagy pathway, enabling cancer cells susceptible to oxidative stress-induced programmed cell death [146]. Inhibition of colon cancer by hesperidin involves multiple alterations, including caspase-3-mediated apoptosis, the autophagy program initiated by PI3K/Akt/GSK-3c and mTOR pathway and, down-regulation of NF-κB and its target molecules iNOS and COX-2 to attenuate oxidative stress and enhance antioxidants to fight tumor-induced inflammation [147,148,149,150]. The above phenomenon also occurred in naringin-treated CRC cells, where the suppression of the PI3K/AKT/mTOR pathway resulted in ameliorated abnormal proliferation and apoptosis [151].

### 5.7. Esophageal Cancer

Esophageal cancer, another intractable disease of the digestive system, is also accompanied by a particularly high fatality rate, with about 1 in every 18 cancer patients dying from it [45]. The way that naringin combats esophageal cancer is to suppress the proliferation and colony formation and the invasion of Eca109 cells by regulating related proteins to block the JAK/STAT signaling pathway, so as to promote cell apoptosis [152]. In nude mice, synephrine has a significant inhibitory effect on ESCC xenografts, and in vitro experiments have observed that it down-regulates Galectin-3 to inactivate the AKT/ERK pathway in ESCC cells [153].

### 5.8. Cervical Cancer

Although cervical cancer is the fourth leading cause of cancer death in women, the most common cancer and the leading cause of cancer death in many countries worldwide, it is considered a preventable cancer [45]. The inhibition of NEU3 activity in HeLa cells and A549 cells mediated by naringin resulted in the accumulation of GM3 ganglioside, which further led to the weakening of EGFR signaling, and finally resulted in cell growth restriction. At the same time, the down-regulation of phosphorylation of EGFR and ERK was also involved in the process [154]. Lin R’s data demonstrated that naringin abolishes Wnt/β-catenin signaling and ultimately triggers cell cycle arrest at G0/G1 phase in Cervical cancer cells, while ER stress-induced cell killing is also a pathway for naringin to act on [155]. Naringin also induces cell cycle arrest in the G2/M phase, inhibits cell growth and induces apoptosis via the NF-κB/COX-2-caspase-1 pathway, thereby exerting its anticancer activity on SiHa cells and HeLa cells [156,157].

### 5.9. Bladder Cancer

The incidence of bladder cancer in men is much higher than women, making it the sixth most common cancer and the ninth cause of cancer death [45]. Apoptosis is an important mechanism of CRP against bladder cancer discovered in current research. Nobiletin-induced apoptosis is accomplished through the regulation of endoplasmic reticulum stress via PERK/elF2α/ATF4/CHOP pathway and PI3K/AKT/mTOR pathway, and its inhibitory effect on BFTC cell growth is positively correlated with concentration range [158]. Anti-tumor formation through apoptosis is also suitable for tangeretin by inducing the release of pro-apoptotic factors such as cytochrome c to form an apoptotic complex with activated caspase-9, thereby initiating the apoptotic response and disrupting mitochondrial function, resulting in BFTC-905 cells being cytotoxic [159].

### 5.10. Other Cancers with High Diagnosis Rate

In addition to the above-mentioned cancers, CRP is quite successful in the treatment of other common cancers. The mechanism of nobiletin against osteosarcoma metastasis is to down-regulate the expression of MMP-2 and MMP-9 via ERK/JNK pathways and inhibit the movement, migration and invasion of U2OS and HOS cells through activation of NF-κB, CREB and SP-1 proteins [160]. Moreover, naringin suppresses the migration and invasion of human chondrosarcoma by up-regulating the expression of miR-126 and downregulating VCAM-1 expression [161]. By inhibiting MAPK and AKT/protein kinase B signaling pathway and downregulating cell cycle-related factors, nobiletin achieves the purpose of combating glioma cell proliferation and migration [162]. Suppression of the cyclin-D/cdc-2 complex formation leads to cell cycle arrest at G2/M arrest, decreased glioblastoma cell growth after tangeretin treatment, increased G2/M phase cells and induces apoptosis [163]. Blocking the MAPKs signaling pathway, downregulating the activity and expression of MMPs, thereby inhibiting the invasion, migration and adhesion of U87 cells, is a non-negligible characteristic of naringin’s anti-metastatic properties [164]. Negative effects on cell proliferation by inhibiting the FAK/cyclin D1 pathway, promoting apoptosis via affecting the FAK/BADs pathway and attenuating cell invasion and metastasis through the FAK/MMPs pathway are another example of naringin in the treatment of glioblastoma cells [165]. Naringin-treated Walker 256 carcinosarcoma rats inhibited tumor growth, down-regulated the expression of IL-6 and TNF-α and significantly prolonged the survival rate without the occurrence of cachexia [166]. Induction of apoptosis by activating caspase-3 and up-regulation of intracellular ROS and blocking cell cycle progression in G2 phase are the main ways that hesperidin reduces the viability of gallbladder cancer cells [167]. Inhibition of MAPK pathway, STAT3 activation and down-regulation of ER signaling pathway in the genome seems to be an important mechanism by which hesperidin induces apoptosis or autophagy in ECC-1 cancer cells [168]. Hesperidin inhibits mesothelioma cell growth by inducing apoptosis by downregulating the mRNA and protein expression levels of Sp1 and its regulatory proteins [169]. Hesperidin triggers apoptosis in lymphocyte lineages in a PPARγ-dependent or PPARγ-independent manner and inactivates NF-κB, which in turn sensitizes Ramos cells to chemotherapeutic agent-induced apoptosis [170]. The mechanism of action of hesperidin in NALM-6 cells is manifested in multiple aspects. It can not only play pro-apoptotic and anti-proliferative effects via PPARγ-dependent and PPARγ-independent pathways, but also affect apoptosis and cytotoxicity through PI3K/AKT/IKK signaling pathway [171,172]. As early as 1998, some scholars have found that hesperidin has an inhibitory effect on 12-O-tetradecanoyl-13-phorbo lactate-induced skin tumor [173]. In vitro, hesperidin affected PD-L1 expression in HN6 cells and HN15 cells by reducing the phosphorylation of STAT1 and STAT3, thereby inhibiting cancer cell survival and avoiding evasion of antitumor immunity. Correspondingly, in vivo experiments found that hesperidin had a negative effect on 4-NQO-induced proliferation of rat oral cancer [174,175,176]. Nobiletin inhibited the proliferation of TCA-8113 cells and CAL-27 cells through cell cycle arrest in G1 phase, accompanied by changes in intracellular levels of acidified PKA and phosphorylated CREB, impaired mitochondrial function, glucose consumption and pyruvate and lactate production [177]. The antagonism of nobiletin on LPS- and INF-γ-induced PGE2, COX-2, NO endows it with the property of preventing inflammation-related tumors [178]. Nobiletin has positive and negative regulatory effects on MMPs and TMP-1, which is a specific manifestation of its interference with PI3K signaling pathway to suppress tumors [179].

## 6. Conclusions

To sum up, the performance of CRP, especially its flavonoids, in the fight against cancer, is worthy of recognition because, while it does not prevent the process of a certain aspect of cancer alone, it reverses or suppresses the development of cancer through various pathways, which is a characteristic that traditional anti-cancer agents lack compared to traditional Chinese medicine. Because of the multiple functions of CRP, its ability has not been thoroughly studied. In addition, CRP is also commendable as an advantage in that it is a natural product that is homologous to medicine and food, which makes it inexpensive and easy to obtain. However, the most prominent defect of CRP as an anti-cancer agent is the lack of clinical research, which will be an important content of CRP to be explored next.

## Figures and Tables

**Figure 1 molecules-27-05622-f001:**
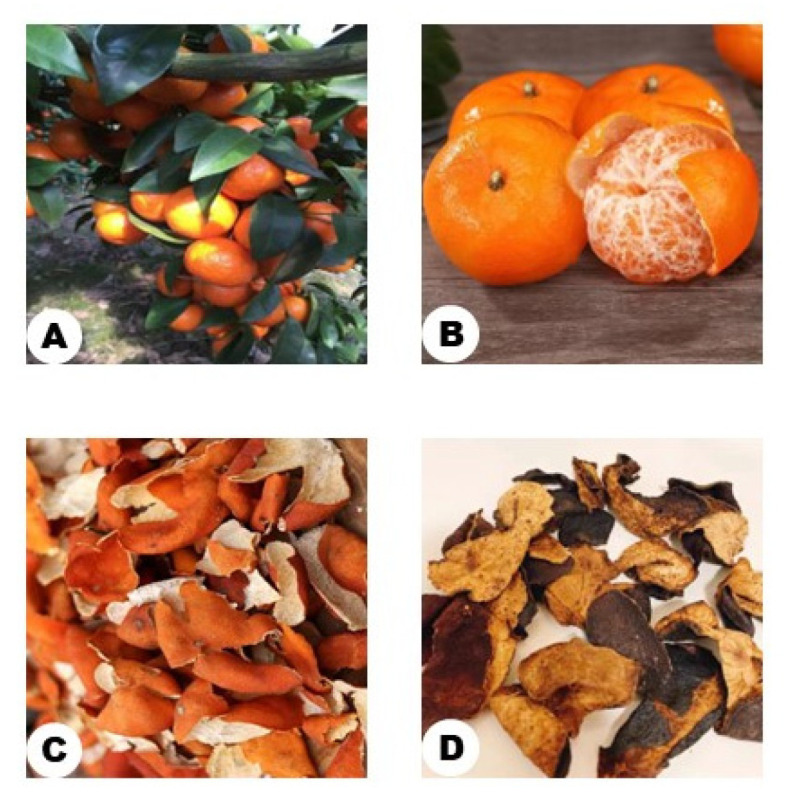
(**A**) Citrus reticulata Blanco tree (**B**) fresh ripe citrus (**C**) fresh mature pericarps (**D**) CRP.

**Figure 2 molecules-27-05622-f002:**
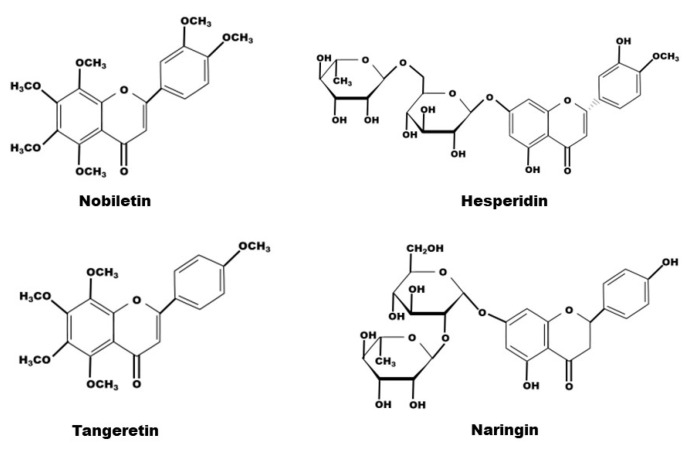
The chemical structure of nobiletin, hesperidin, tangeretin and naringin contained in CRP.

**Figure 3 molecules-27-05622-f003:**
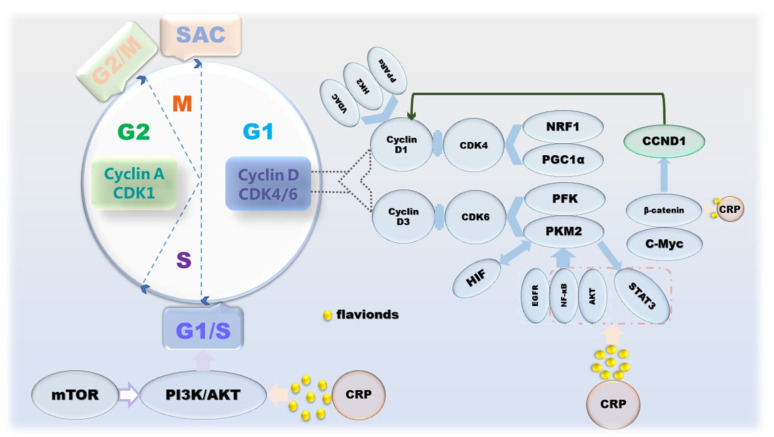
The mechanism of CRP causing cancer cell cycle arrest.

**Figure 4 molecules-27-05622-f004:**
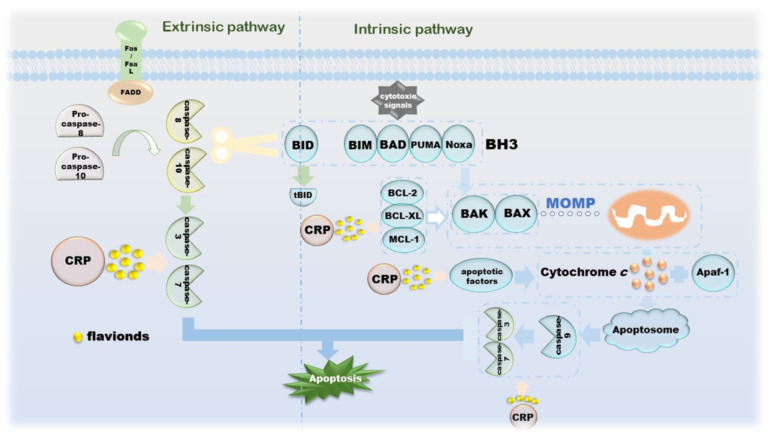
The mechanism of CRP promoting apoptosis of cancer cells.

**Figure 5 molecules-27-05622-f005:**
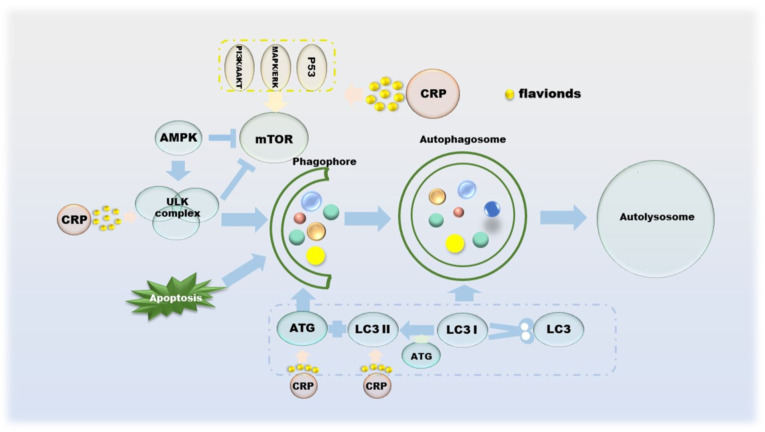
The role of CRP in the process of autophagy in cancer cells.

**Figure 6 molecules-27-05622-f006:**
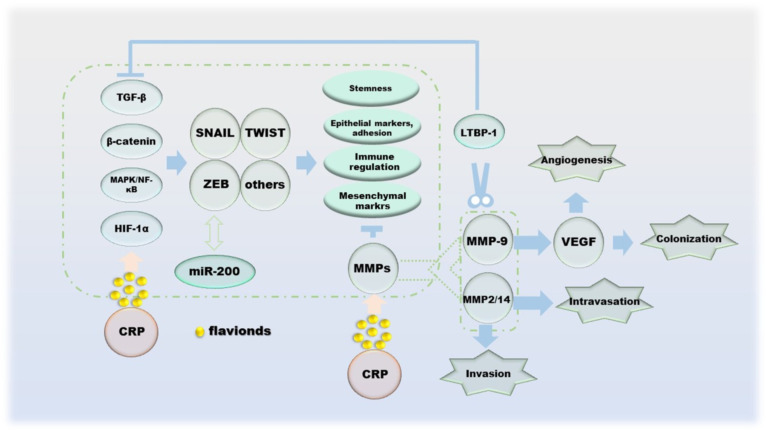
The effect of CRP during the migration of cancer cells.

**Table 1 molecules-27-05622-t001:** The Flavonoids in CRP.

Classification	Chemical Component	Molecular Formula	Molecular Mass	References
Polymethoxyflavones	Nobiletin	C_21_H_22_O_8_	402	[40]
	3,5,6,7,8,3′,4′-Heptamethoxyflavone	C_22_H_24_O_9_	433	[40]
	5-Hydroxy-6,7,8,3′,4′-pentamethoxyflavone	C_20_H_20_O_8_	388	[40]
	Tangeretin	C_20_H_20_O_7_	373	[40]
	Monohydroxy-trimethoxyflavone	C_18_H_16_O_6_	329	[40]
	Monohydroxy-tetramethoxyflavone	C_19_H_18_O_7_	359	[40]
	Monohydroxy-pentamethoxyflavone	C_20_H_20_O_8_	389	[40]
	Trihydroxy-dimethoxyflavone	C_17_H_14_O_7_	331	[40]
	Trihydroxy-trimethoxyflavone	C_18_H_16_O_8_	361	[40]
	Isosinensetin	C_20_H_20_O_7_	373	[40]
	Monohydroxy-hexamethoxyflavone	C_21_H_22_O_9_	419	[40]
	Tetramethoxyflavone	C_19_H_18_O_6_	343	[40]
	Hexamethoxyflavone	C_21_H_22_O_8_	403	[40]
	Sinensetin	C_20_H_20_O_7_	373	[40]
	Tetramethyl-O-isoscutellarein	C_19_H_18_O_6_	343	[40]
	Dihydroxy-trimethoxyflavone	C_18_H_16_O_7_	345	[40]
	Trimethoxyflavone	C_18_H_16_O_5_	313	[40]
	Pentamethoxyflavone	C_20_H_20_O_7_	373	[40]
	Tetramethyl-O-scutellarein	C_19_H_18_O_6_	343	[40]
	Dihydroxy-tetramethoxyflavone	C_19_H_18_O_8_	375	[40]
	Dihydroxy-pentmethpxyflavone	C_20_H_20_O_9_	405	[39]
	Natsudaidai	C_19_H_18_O_7_	359	[40]
Flavanone O-glycosides	Hesperidin	C_28_H_34_O_15_	611	[40]
	Naringin	C_27_H_32_O_14_	581	[40]
	Eriocitrin	C_27_H_32_O_15_	597	[40]
	Neohesperidin	C_27_H_32_O_15_	597	[40]
	Narirutin	C_27_H_32_O_14_	581	[40]
	Prunin	C_21_H_22_O_10_	435	[40]
	Neohesperidin	C_28_H_34_O_15_	611	[40]
	Poncirin	C_28_H_32_O4_14_	595	[40]
	Didymin	C_28_H_34_O_14_	595	[40]
	Melitidin	C_33_H_40_O_18_	725	[40]
Flavone O-glycosides	Rhoifolin	C_27_H_30_O_14_	579	[40]
	Hexamethoxyflavone-o-glucoside	C_27_H_33_O_13_	565	[40]
	Luteolin-7-O-rutinoside	C_27_H_30_O_15_	595	[40]
	Diosmin	C_28_H_32_O_15_	609	[40]
	Neodiosmin	C_28_H_32_O_15_	609	[40]
	Sudachiin C or B	C_30_H_34_O_17_	667	[40]
Flavone C-glycosides	Vicenin-2	C_27_H_30_O_15_	595	[40]
	Diosmetin-6,8-di-C-glucoside	C_28_H_32_O_16_	625	[40]
	Lucenin-2	C_27_H_30_O_16_	611	[40]
	Apigenin-8-C-glucoside	C_21_H_20_O_10_	433	[40]
	Diosmetin-6-C-glucoside	C_22_H_22_O_11_	463	[40]
	Apigenin-6,8-di-C-glucoside	C_27_H_30_O_2_	595	[39]
	Chysoeriol-6,8-di-C-glucoside	C_28_H_32_O_16_	625	[39]
Flavanone aglycones	Naringenin	C_15_H_13_O_5_	273	[40]
	Hesperetin	C_16_H_14_O_6_	303	[40]

## Data Availability

Not applicable.

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
