# Peer review of "Mechanism of Citri Reticulatae Pericarpium as an Anticancer Agent from the Perspective of Flavonoids: A Review"

_molecules, 2022, doi:10.3390/molecules27175622_

Round 1
Reviewer 1 Report
Review report for the manuscript:
“Mechanistic of Citri Reticulatae pericarpium as an anticancer agent from the perspective of flavonoids: A review”
The manuscript with title “Mechanistic of Citri Reticulatae pericarpium as an anticancer agent from the perspective of flavonoids: A review” (ID-1865796) discusses in detail the impact and significance of Citri reticulatae pericarpium against cancer and in parallel exposes in detail the existing data of the exact mechanism of action in the most important cancer types. In general, the manuscript falls within the scope of Molecules research topic, section “Natural products Chemistry” and special issue “A feasible Approach for Natural products to treatment of Diseases“.
Overall, the manuscript is well written and gather important data for the significance of the studied material in the fight of cancer. The existing data are presented in detail and cover a significant bibliographical overview in a comprehensible manner.
However, the manuscript has few mistakes (grammatik and syntactic) and English editing is required before publication. Also major and minor revisions are proposed before publication to improve the quality of the manuscript. Below find in detail the minor and major revisions:
Minor revisions
General
1. Title. Maybe the use of the word “mechanism” instead of “mechanistic” it is more correct verbally.
2. Keywords. Please add the complete meaning of CRP and not the abbreviation.
3. Italicized “in vitro” in the whole text.
4. When you refer to a research study and you refer to the first author add “et. al.” after the first name (i.g Chen M. et. al., Zheng J. et al.) and correct it in the whole manuscript.
Introduction
1. Page 1, line 1: Explain the CRP. It is the first time reported in the main text
2. Page 1, line 2: Citrus reticulatae should be italicized
3. Page 1 line 6: “to form a compound”. Please rephrase because it is not clear the meaning of the sentence
Pharmacology pf CRP
1. Title. Improve the title, because it is not descriptive and it is not in accordance with the content of the paragraph. The paragraph contains pharmacological data along with information regarding CRP chemical composition.
2. Page 2 line 4: Please rephrase. The meaning of the sentence is not clear. Also based on the cited bibliography the positive affect against neurodegenerative diseases is attributed to the pure compounds (flavonoids) and to CRP extract, you should clarify it.
3. Caption of figure 2. Please rephrase. There are grammatical errors and the caption should be fully explanatory.
Cancer
1. Title. Improve the title. It is very general and not descriptive.
2. Page4, line 18. Define TCM.
3. Page 4, line 21-26. These sentences should be moved at the end of introduction (explained in detail in the major revisions).
Major revisions
1. In the introduction, an overview about the pharmacological and especially the anti-cancer effects of CRP is presented. However, the impact of the current review to the existing knowledge in not described and so the novelty and significance of the review is not clear to the reader. It would be more appropriate to add a paragraph at the end of introduction explaining the concept of the review, the kind of data that are discussed, the significance of the data, the way that are presented and the difference/novelty of the current review in comparison to other similar works.
2. In the whole text, you refer to pathways, specific signalings and proteins without explain their role in the cell cycle and cancer progression. It would be more explanatory to move paragraph 5 “The performance of CRP in the typical phenotype of cancer” before the quotation of the impact of CRP in the specific type of cancers (breast cancer, lung cancer etc.). By this way you will explain the exact mechanism of action of CRP and you will have fully explain the signaling, pathways and the production of the associated proteins before presenting the data, making the text more descriptive.
Author Response
Dear Reviewer:
I am very grateful for your comments on this manuscript. we have carefully studied your comments and revised the manuscript according to your advices. Below are my point-by-point responses to your questions and comments.
General
Point 1: Title. Maybe the use of the word “mechanism” instead of “mechanistic” it is more correct verbally.
Response 1: Thanks to Reviewer for the correction, we have changed “mechanistic” to “mechanism”. (in red)
Point 2: Keywords. Please add the complete meaning of CRP and not the abbreviation.
Response 2: We have added the full meaning of CRP, Citri Reticulatae Pericarpium, in the Keywords section. (in red)
Point 3: Italicized “in vitro” in the whole text.
Response 3: Thanks to Reviewer for reminder, we have italicized "in vivo" in the whole text. (in red)
Point 4: When you refer to a research study and you refer to the first author add “et. al.” after the first name (i.g Chen M. et. al., Zheng J. et al.) and correct it in the whole manuscript.
Response 4: Thanks to Reviewer for reminder, we have revised the author's formatting throughout the manuscript in accordance with your comments. (in red)
Introduction
Point 1: Page 1, line 1: Explain the CRP. It is the first time reported in the main text.
Response 1: Following Reviewer’s comments, in the revised manuscript, we have explained the definition of CRP in the first mention of it. (in red)
Point 2: Page 1, line 2: Citrus reticulatae should be italicized.
Response 2: Thanks to Reviewer for reminder, we have italicized " Citrus reticulatae " in the revised manuscript. (in red)
Point 3: Page 1 line 6: “to form a compound”. Please rephrase because it is not clear the meaning of the sentence.
Response 3: Based on your comments, we have rewritten this sentence as " form many well-known classical prescriptions." (in red)
Pharmacology pf CRP
Point 1: Title. Improve the title, because it is not descriptive and it is not in accordance with the content of the paragraph. The paragraph contains pharmacological data along with information regarding CRP chemical composition.
Response 1: We have adopted your comments and rephrased the title to " Pharmacological effects and chemical composition of CRP ".(in red)
Point 2: Page 2 line 4: Please rephrase. The meaning of the sentence is not clear. Also based on the cited bibliography the positive affect against neurodegenerative diseases is attributed to the pure compounds (flavonoids) and to CRP extract, you should clarify it.
Response 2: Sorry we missed something crucial, the full sentence should be " Experiments have found that CRP’s neuroprotective properties confer the ability to prevent neurodegeneration caused by Parkinson’s, Alzheimer’s, Huntington’s disease and multiple sclerosis. "(in red)
Point 3: Caption of figure 2. Please rephrase. There are grammatical errors and the caption should be fully explanatory.
Response 3: We are very sorry that the title of Figure 2 could not be explained clearly, and it has now been revised and written as " The chemical structure of nobiletin, hesperidin, tangeretin and naringin contained in CRP." (in red)
Cancer
Point 1: Title. Improve the title. It is very general and not descriptive.
Response 1: Thanks for the suggestion, we didn't have a strong connection between the title and the content before. The content of this paragraph is the epidemiological information of cancer, and the title is now changed to "Epidemiological investigation of cancer".(in red)
Point 2: Page4, line 18. Define TCM.
Response 2: TCM is defined as Traditional Chinese Medicine, we have made corrections in the manuscript. (in red)
Point 3: Page 4, line 21-26. These sentences should be moved at the end of introduction (explained in detail in the major revisions).
Response 3: At your suggestion, we have moved this paragraph to the end of the introduction. (in red)
Major revisions
Point 1: In the introduction, an overview about the pharmacological and especially the anti-cancer effects of CRP is presented. However, the impact of the current review to the existing knowledge in not described and so the novelty and significance of the review is not clear to the reader. It would be more appropriate to add a paragraph at the end of introduction explaining the concept of the review, the kind of data that are discussed, the significance of the data, the way that are presented and the difference/novelty of the current review in comparison to other similar works.
Response 1: As you mentioned, adding the concept and data of the review at the end of the introduction is more intuitive to reflect the novelty of the current review, and it can also deepen the concept of CPR against cancer. At the end of the introduction, we have moved the content of lines 21-26 of the previous version of the review here to illustrate the anticancer mechanism of CRP and flavonoids and experimental data in combating various cancers. (in red)
Point 2: In the whole text, you refer to pathways, specific signalings and proteins without explain their role in the cell cycle and cancer progression. It would be more explanatory to move paragraph 5 “The performance of CRP in the typical phenotype of cancer” before the quotation of the impact of CRP in the specific type of cancers (breast cancer, lung cancer etc.). By this way you will explain the exact mechanism of action of CRP and you will have fully explain the signaling, pathways and the production of the associated proteins before presenting the data, making the text more descriptive.
Response 2: In the earliest designs, we originally intended to put the exact mechanism of action of CRP against cancer ahead of the experimental data, but the current model was adopted for various reasons. After reading your comments, we also believe that moving "The performance of CRP in the typical phenotype of cancer" to the front makes the impact of CRP in specific types of cancer more understandable. (in red)
Reviewer 2 Report
This manuscript reviews the anticancer mechanisms of several flavonoids contained in CRP, including nobiletin, tangeretin, naringin, and hesperidin, to evaluate the anticancer potential of CRP. Overall, the manuscript is interesting and acceptable, but there are still some errors and issues that need to be corrected. My specific opinions are as follows:
1. The author should carefully check the word or sentence and correct spelling and grammatical errors, such as:originat → originate.
2. It would be better to add 2-3 keywords.
3. Please provide the full name of the abbreviations when firstly mentioned, such as BCSCs, HCC, etc. Please check the complete manuscript carefully.
4. Note the summary section. The advantages and disadvantages of CRP as an anticancer agent are less elaborated, please add further discussion.
Author Response
Dear Reviewer:
I am very grateful for your comments on this manuscript. we have carefully studied your comments and revised the manuscript according to your advices. Below are my point-by-point responses to your questions and comments.
Point 1: The author should carefully check the word or sentence and correct spelling and grammatical errors, such as: originat → originate.
Response 1: Thank you for your reminder, we have checked the spelling of the full text and corrected the wrong words. (in red)
Point 2: It would be better to add 2-3 keywords.
Response 2: At your suggestion, we have added two new keywords, "mechanism" and "phenotype".(in red)
Point 3: Please provide the full name of the abbreviations when firstly mentioned, such as BCSCs, HCC, etc. Please check the complete manuscript carefully.
Response 3: We checked the manuscript and added the full name where its corresponding abbreviation first appeared. (in red)
Point 4: Note the summary section. The advantages and disadvantages of CRP as an anticancer agent are less elaborated, please add further discussion.
Response 4: The last paragraph of the manuscript summarizes the advantages and disadvantages of CRP as an anticancer agent, and it is clear that the original manuscript has shortcomings. At your suggestion and after careful consideration, we have added the source and economic advantages of CRP over traditional anticancer agents. (in red)
Round 2
Reviewer 1 Report
Thank you for the collaboration and the revisions. The manuscript is ok for publication.